# U-Pb dating on calcite paleosol nodules: first absolute age constraints on the Miocene continental succession of the Paris Basin

Vincent Monchal[1], Rémi Rateau[1], Kerstin Drost[1], Cyril Gagnaison[2], Bastien Mennecart[3], Renaud Toullec[2], Koen Torremans[4], David Chew[1]

[1]Geology, School of Natural Sciences, Trinity College Dublin, Dublin, D02 PN40, Ireland

[2]Département Géosciences, Unité Bassins-Réservoirs-Ressources (B2R-U2R 7511), Institut Polytechnique UniLaSalle Beauvais, UniLaSalle-Université de Picardie, Beauvais, 30313, France

[3]Naturhistorisches Museum Basel, Basel, 4001, Switzerland

[4] School of Earth Sciences, University College Dublin, Belfield, Dublin 4, Ireland

*Correspondence to*: Vincent Monchal (monchalv@tcd.ie)

## Abstract

Continental sedimentary successions are typically less complete and more poorly preserved than the marine record, leading to limited correlations between basins. Traditionally, intra-basin correlation employs radiometric dating of volcanic markers or relative dating based on the fossil record. However, volcanic markers may not always be present, and biostratigraphy relies on index fossils that are often sparse to absent in continental succession. Recent progress in carbonate U-Pb dating can improve correlations between continental successions by providing absolute age constraints on carbonate deposition and/or on syn- to post-depositional processes such as pedogenesis.

In this study, we analysed pedogenic calcite nodules within a continental Miocene succession in the southwestern Paris Basin (the important paleontological site at Mauvières quarry, France). Following multimethod petrographic characterisation of the samples, LA-ICP-MS U-Pb dating was employed to obtain formation ages on the pedogenic calcite nodules. The Tera-Wasserburg intercept ages from five nodules from the same horizon (19.3±1.3/1.4 Ma, 18.8±2.7/2.7 Ma, 19.11±0.84/0.94 Ma, 19.0±2.3/2.3 Ma and 19.4±2.7/2.7 Ma) are in excellent agreement with previous biostratigraphic constraints on the sequence. Petrographic evidence points to a single crystallisation event, and we conclude that the formation of the calcite nodules occurred at 19.22 ± 0.66/0.79 Ma (central age from a radial plot of the five Tera-Wasserburg intercept ages). This calcite formation age is regarded to represent a minimum depositional age of the strata hosting the root nodules. It provides the first absolute age for the continental Miocene succession (and Neogene mammal zone MN3) of the Paris Basin and allows correlation with other continental basins independent of their fossil assemblages or where fossil content is absent.

## 1 Introduction

Biostratigraphy assigns relative ages to rock strata by using the fossil assemblages contained within them, with the goal of showing that a particular horizon in a given section represents a similar period of time as an analogue horizon in a different succession. It relies heavily on the presence of index fossils - fossils with a limited time range, wide geographic distribution, and rapid evolutionary trends. The common absence of biostratigraphically-diagnostic index fossils in continental successions is problematic, and absolute dating approaches often need to be applied to continental successions. Such approaches include geochronology of volcanic horizons such as lava flows, ash beds, or cryptotephra (*e.g.*, Rubidge et al., 2013; Smith et al., 2017; Poujol et al., 2023), astronomic calibration (e.g., Kerr 1992, Montano et al., 2021), and magnetostratigraphic correlation (e.g., Kalin and Kempf 2009). While volcanic horizons can provide accurate and precise absolute ages, they are not ubiquitous in the sedimentary record. Carbonates are very common in terrestrial successions (except in humid climates) where they can be classified as pedogenic or non-pedogenic, depending on whether they have formed by soil-forming processes (Zamanian et al., 2016). Pedogenic carbonates comprise calcretes and dolocretes - paleosols that have been indurated by a pervasive calcitic cement; pisoliths - globular nodules made of concentric calcitic spheres; and more generic calcitic nodules - indurated concretions with a globular or cylindrical shape, often associated with calcitic cementation around plant roots (rhizocretions; Zamanian et al., 2016).

The formation of carbonates nodules can be classified according to the morphology of the nodule and the postulated fluid pathway that led to the formation of the nodule (Zamanian et al., 2016). *Perdescendum* models and *Perascendum* models involve dissolution of carbonate with reprecipitation in a different horizon (a deeper horizon for *Perdescendum* and shallower for *Perascendum*) while *in situ* models do not imply significant carbonate migration through the soil profile (Zamanian et al., 2016). Biological models invoke absorption of Ca-enriched fluid by an organism, leading to calcification of Ca-bearing organs or supersaturation that induces carbonate precipitation (Zamanian et al., 2016). These biological models include rhizolith formation, whereby plant roots pump the water from Ca-enriched fluids leaving behind residual $Ca^{2+}$ ions that react with the $CO_2$ emitted by rhizomicrobial respiration, resulting in the earliest carbonate cements around the root (Zamanian et al., 2016). After the root dies, the void created is filled (partially to completely) by calcite resulting from the activity of bacteria, algae, or by dissolution of the early carbonate cement and reprecipitation into cavities (Aguirre Palafox et al., 2024). When compaction starts, intergranular space and compaction cracks can create new cavities for carbonate precipitation. With burial, the nodule can travel from the oxidising conditions of the vadose zone towards the more reducing environment of the phreatic zone (Aguirre Palafox et al., 2024). This results in chemical changes (e.g. in Fe, Mn, and Pb) observable in cathodoluminescence (CL) images but which also affect U-Pb geochronology (Aguirre Palafox et al., 2024).

U-Pb dating of calcium carbonate started in the late 1980s using isotope dilution (ID) – thermal ionisation mass spectrometry (TIMS) methods (Smith and Farquhar, 1989; Roberts et al., 2020 and references therein). Most terrestrial U-Pb carbonate dating studies have focused on non-pedogenic carbonates (e.g. speleothems, tufas, and lacustrine carbonates; see review in Rasbury and Cole, 2009 and a more recent review by Rasbury et al., 2023). The first U-Pb dating studies applied to terrestrial

pedegenic carbonates took place in the mid-1990s on Paleozoic uranium-rich dolocretes, developed subaerially on top of marine limestones (Hoff et al., 1995; Winter and Johnson, 1995). Over the following years, a series of ID-TIMS U-Pb dating studies yielded meaningful subaerial exposure ages from 1) Late Paleozoic paleosol-derived sparry calcite developed on top of marine carbonate cyclothems in the southwestern USA (Rasbury et al., 1997, 1998, 2000; Rasbury and Cole, 2009); 2) Late Paleozoic dolocretes from Kansas, USA (Luczaj and Goldstein, 2000); 3) Late Paleozoic subaerially soil-modified palustrine limestones in Ohio, USA (Becker et al., 2001); and 4) Triassic calcretes developed on top of fluviatile siliciclastics deposits in Connecticut, USA (Wang et al., 1998) (Table 1).

Since the mid-2010s, advances in laser ablation inductively coupled plasma mass spectrometry (LA-ICP-MS) have allowed U-Pb dating of carbonates with the benefits of much greater spatial resolution, mineralogical context (being in-situ), and sample throughput (e.g., Li et al., 2014; Roberts and Walker, 2016; Nuriel et al., 2017). While individual LA-ICP-MS spot ablation U-Pb data are typically significantly less precise than ID-TIMS U-Pb analyses, the high spatial resolution of the approach means it can commonly encounter both high and low U/Pb portions of the sample, resulting in age regressions with superior precision compared to ID-TIMS U-Pb dating studies which employ bulk sampling (e.g., Li et al., 2014; Roberts et al., 2020). The technique has been employed to date pedogenesis including 1) Eocene pedogenic calcite nodules from Montana, USA (Methner et al., 2016), 2) an Upper Triassic continental succession with calcite nodules and interbedded volcanic markers from Argentina (Aguirre Palafox et al., 2024), 3) Ediacaran dolomite from subaerially weathered volcanics in Ukraine (Liivamägi et al., 2021) (Table 1), and 4) U-Pb geochronology on *Microcodium* calcite from the Spanish Southern Pyrenees that provided more constraints on fluvial mobility during the Paleocene–Eocene Thermal Maximum event (Prieur et al., 2024). LA-ICP-MS U-Pb dating requires chemically homogenous and large enough zones (typically, between 50 and 200 μm wide circles or squares) to obtain sufficient U and radiogenic Pb signals to produce meaningful age results (Roberts and Holdsworth, 2022). Additionally, high U and low common Pb concentrations are required to produce precise U-Pb dates, but carbonates typically incorporate low U abundances (unless the precipitation takes place in reducing environments, e.g Fournier et al., 2004; Drake et al., 2018; Aguirre Palafox et al., 2024) and significant common Pb (Roberts et al., 2020). Carbonates in general and pedogenic carbonates in particular, are also often heterogenous at the hundreds of micrometre scale or below (Zamanian et al., 2016; Roberts and Holdsworth, 2022; Aguirre Palafox et al., 2024), partially explaining the paucity of reliable dating results from pedogenic carbonates. More widespread absolute dating of pedogenic carbonates may provide valuable chronostratigraphic constraints in continental successions, particularly those where volcanic horizons or index fossils are absent. Aguirre Palafox et al. (2024) recently provided guidelines and strategies to improve the sampling and interpretation of pedogenic carbonates, and addressed the influence of redox conditions on U concentrations and potential internal zonation.

Table 1: Summary of published U-Pb ages of terrestrial pedogenic carbonates (modified and updated after Rasbury and Cole, 2009).

| Age | ±2σ | MSWD | Max. U | Technique | Material dated | | | Soil protolith | Reference |
|---|---|---|---|---|---|---|---|---|---|
| Ma | Ma | - | ppm | | Country | Rock | Mineral | | |
| 39.5 | 1.4 | 0.89 | 3.25 | LA-SF-ICP-MS (spots) | USA | Pedogenic nodule | Cal | clastics & volcanics - continental | Methner et al., 2016 |
| 40.1 | 0.8 | 1.15 | 3.44 | LA-SF-ICP-MS (spots) | USA | Pedogenic nodule | Cal | clastics & volcanics - continental | Methner et al., 2016 |
| 52.9 | 15 | 4.1 | ? | LA-SF-ICP-MS (spots) | Spain | Microcodium | Cal | sandstones - fluviatile | Prieur et al., 2024 |
| 72 | 11 | 0.011 | ? | LA-SF-ICP-MS (spots) | Spain | Microcodium | Cal | sandstones - fluviatile | Prieur et al., 2024 |
| 80.9 | 11 | 30 | 0.6 | ID-TIMS | USA | Rhizolith | Cal, blocky | clastics - fluviatile | Wang et al., 1998 |
| 211.9 | 2.1 | 2.67 | 2.7 | ID-TIMS | USA | Calcrete | Cal, micritic | clastics - fluviatile | Wang et al., 1998 |
| 212.4 | 3.4 | 3.4 | 2.5 | ID-TIMS | USA | Calcrete | Cal, micritic | clastics - fluviatile | Wang et al., 1998 |
| 228.4 | 5 | 1.7 | 7 | LA-SF-ICP-MS (spots) | Argentina | Pedogenic nodule | Cal | clastics & volcanics - fluviatile | Aguirre Palafox et al., 2024 |
| 230.5 | 2.2 | 1.1 | 40 | LA-SF-ICP-MS (spots) | Argentina | Pedogenic nodule | Cal | clastics & volcanics - fluviatile | Aguirre Palafox et al., 2024 |
| 233.6 | 3.9 | 0.89 | 120 | LA-SF-ICP-MS (spots) | Argentina | Pedogenic nodule | Cal | clastics & volcanics - fluviatile | Aguirre Palafox et al., 2024 |
| 254 | 29 | 504 | 29 | ID-TIMS | USA | Dolocrete | Dol | carbonates - marine | Luczaj and Goldstein, 2000 |
| 275 | 6 | ? | ? | ID-TIMS | USA | Paleosol | Cal | carbonates - lacustrine? | Becker et al., 2001 |
| 282 | 28 | 417 | 32.5 | ID-TIMS | USA | Dolocrete | Dol | carbonates - marine | Hoff et al., 1995 |
| 294 | 6 | ? | ? | ID-TIMS | USA | Paleosol | Cal | carbonates - lacustrine? | Becker et al., 2001 |
| 294.9 | 8.6 | 2.2 | ~27 | LA-Q-ICP-MS (map) | USA | Calcrete | Cal, sparry | carbonates - marine | Rasbury et al., 2023 |
| 298.1 | 1.4 | 0.9 | 8.6 | ID-TIMS | USA | Calcrete | Cal, sparry | carbonates - marine | Rasbury et al., 1997, 1998, 2000, 2009 |
| 306 | 2.6 | 0.6 | - | ID-TIMS | USA | Calcrete | Cal, sparry | carbonates - marine | Rasbury et al., 1998 |
| 512 | 10 | 314 | 1.24 | ID-TIMS | USA | Dolocrete | Dol | carbonates - marine | Winter and Johnson, 1995 |
| 548 | 19 | 1.3 | 0.57 | LA-SF-ICP-MS (spots) | Ukraine | Weathered volcanics | Dol, blocky | volcanics - basalts, tuffs | Liivamägi et al., 2018 |

Abbreviations: Cal = Calcite, Dol = Dolomite, ID-TIMS = Isotope Dilution Thermal Ionization Mass Spectrometer, LA-(SF-/Q-)ICP-MS = Laser Ablation (Sector Field/Quadrupole) Inductively Coupled Mass Spectrometry

A recent and innovative LA-ICP-MS U-Pb carbonate dating protocol, based on the selection and pooling of pixels from 2D elemental and isotopic ratio maps (Drost et al., 2018; Roberts et al., 2020; Chew et al., 2021) is now commonly employed as a U-Pb dating strategy (e.g. Monchal et al., 2023; Rasbury et al., 2023; Subarkah et al., 2024). This in-situ technique allows for the selection of chemically homogenous zones within a chemically heterogenous ablated 2D map area, reducing the risk of incorporating U-Pb data from non-carbonate inclusions or different generations of carbonates (Drost et al., 2018). In addition, this method optimises the spread of data points in Tera-Wasserburg (TW) space increasing the precision of the results (Drost et al., 2018). Therefore, this mapping-based technique is well suited to U-Pb dating and elemental characterisation of paleosol calcite, and can help alleviate some of the issues caused by microheterogeneity in pedogenic carbonates. A late Paleozoic paleosol calcite, already dated by ID-TIMS (298.1±1.4 Ma; Rasbury et al., 1998) has been successfully dated using this approach (294.9±8.6 Ma; Rasbury et al., 2023).

Continental sedimentary successions are often barren or poor in index fossils, which makes dating and intra-basin correlation difficult. Mammal remains have been used to create terrestrial biostratigraphic scales, such as the Neogene mammal (MN) scale in Western Europe (Mein, 1975; Agustí et al., 2001). The European MN scale is similar to the North American Land Mammal Ages (NALMAs) and South American Land Mammal Ages (SALMAs) scales (see the review of Hilgen et al., 2012). However, mammal fossils are not ubiquitous in the sedimentary record, thus the MN and other mammal scales cannot always be employed. When compared to marine biostratigraphic records (which have index fossils such as planktonic foraminifera, ammonites, graptolites, etc.), they also exhibit diachronicity and a poorer temporal resolution. The poor temporal resolution, particularly in the early stages of the MN scale (see Discussion) is well illustrated by the MN3 biozone, which has a duration of between 2.8 and 5.4 Myr, depending on the absolute age chosen for the top and bottom boundaries (Mein, 1999; Steininger, 1999; Aguilar et al. 2003; Raffi et al., 2020). For comparison, the Paleogene calcareous nannofossil scale biozones all have a duration lower than 2 Myr, with most lower than 1 Myr (Agnini et al., 2014). The LA-ICP-MS calcite U-Pb geochronology

approach adopted in this study has the potential to constrain the age of continental sedimentary horizons where pedogenic nodules are present. This approach may improve inter-basin correlations, temporal resolution of the MN scale, as well as potentially highlighting regional diachronism if more extensive sampling campaigns were conducted. In this study, we apply the LA-ICP-MS U-Pb mapping technique along with spot analysis to obtain absolute ages from pedogenic calcite nodules from a terrestrial Miocene succession in the Paris Basin, France, whose age is hitherto poorly constrained in terms of absolute dating.

## 2 Geological setting

### 2.1 Regional geology

The Mauvières paleontological site is located in the SW of the Paris Basin (France), a Mesozoic-Cenozoic intracontinental sag basin (Guillocheau et al., 2000). The site is on the northeast margin of the Neogene outcrops, which comprise continental and marine sedimentary rocks unconformably deposited on Paleogene continental sedimentary rocks (Figure 1a). Regionally, the Cenozoic sedimentary sequence reflects a dominantly continental paleoenvironment with occasional marine transgressions during the Miocene (Figure 1b, Gagnaison, 2020).

### 2.2 Paleoenvironment and origin of the calcite nodules

The pedogenic nodules were sampled from the Early Miocene (early to middle Burdigalian) Orléanais Marls and Sands Formation (Figure 1b), a few meters-thick succession of coarse and fine-grained clastic sediments (Figure 2). This Formation rests unconformably over a Paleogene lacustrine silty marl (the Eocene-Oligocene Grey Marls Formation) and is overlain by Middle Miocene marine shelly carbonate sands, known locally as "*faluns*" (Gagnaison et al., 2023) (Figure 1b and Figure 2). The Early Miocene continental sequence at Mauvières consists of a series of eight clastic beds (numbered s1 to s8; Gagnaison et al., 2023; Figure 2a). The pedogenic nodules were found in the basal bed s1, which overlies Eocene-Oligocene silty marls (Figure 2). The s1 bed is comprised of a very coarse light grey-orangey quarzitic sand with minor feldspar, in-situ terrestrial vertebrate fossils, poorly preserved *Unio* shells (a freshwater mussel), and in-situ carbonate nodules and cylinders. The sand also contains reworked material, including Cretaceous calcareous and siliceous pebbles, altered glauconite grains, and Cretaceous and Oligocene / Early Miocene vertebrate fossils. The sand is loosely cemented with a clayey and calcareous matrix (Gagnaison et al., 2023). Some rare *Unio* shells have been found with both valves still connected, indicating both a low-energy environment and that they are in-situ (i.e. not reworked from older beds).

The occurrence of 1) hollow calcite cylinders and nodules interpreted as rhizocretions, 2) root tracks in the matrix, 3) iron oxides that give the sand its orange colour, and 4) microvacuoles interpreted as products of subaerial microbial activity, suggest the presence of a paleosol (Gagnaison et al., 2023). The nodule-bearing s1 bed is interpreted as a water-transported, low-energy fluvial sequence with prograding sand bars, with phases of lacustrine floodings and development of paleosols. The sequence was subsequently subaerially exposed and followed by the development of a vegetated soil (Gagnaison et al., 2023). The

pedogenic nodules are consequently interpreted as in-situ and not reworked from older horizons. Rasbury et al. (1997) have
shown that the calcite spar typically forms rapidly following paleosol development, and therefore absolute dating of the
nodules should provide robust age constraints on the minimum age of the s1 bed. Based on detailed petrography including CL
imaging, Aguirre Palafox et al. (2024) provided more information on the environmental factors that influence the timing of
nodule formation (e.g. redox conditions, burial, water table levels) that can in turn help refine the interpretation of
geochronological results.

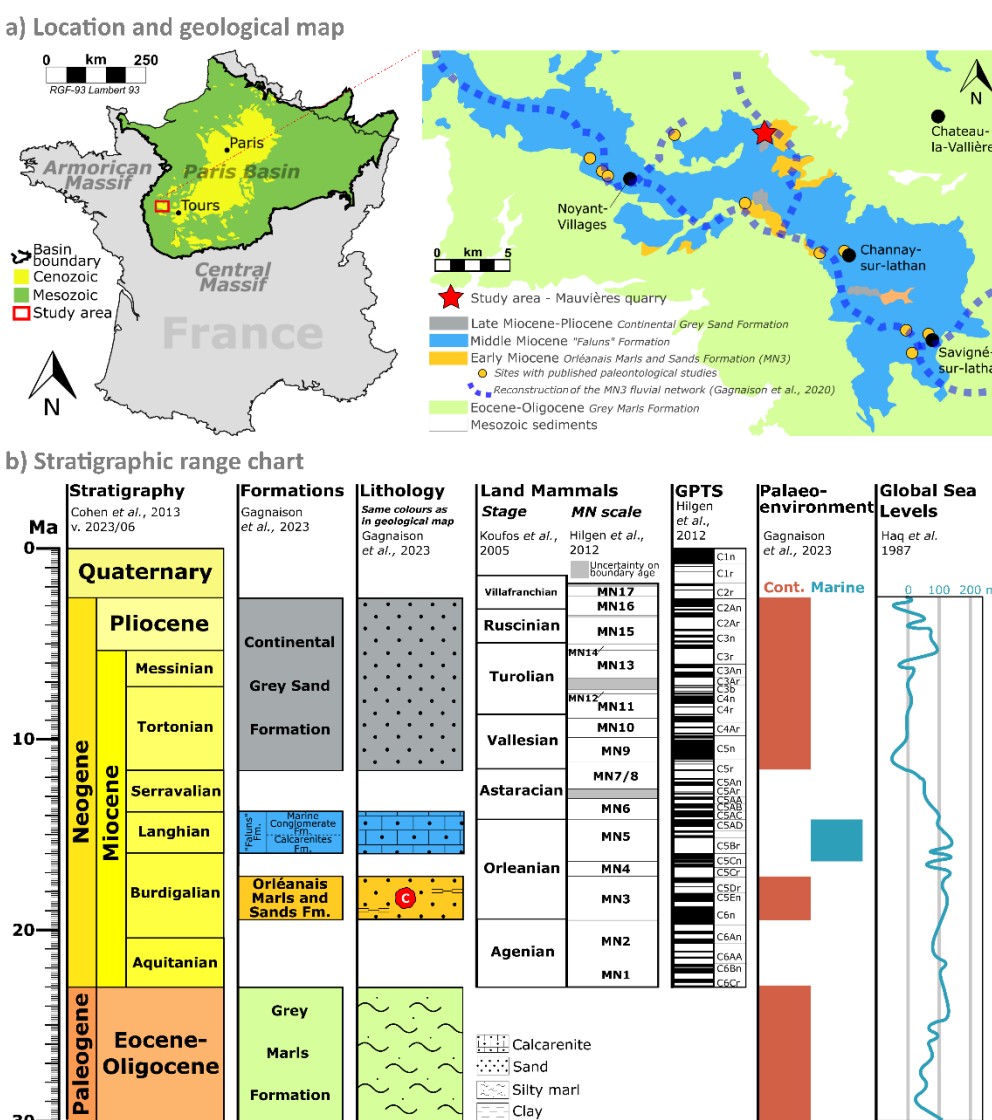

**Figure 1: Geological context of the Mauvières section. a) Location of the Mauvières quarry and regional geology based on the**
**BRGM 1/50,000 unified vector geological map of France (InfoTerre), modified after Gagnaison et al. (2020). b) Stratigraphy of the**
**Mauvières section. The nodules (red symbol with a C) come from the Orléanais Marls and Sands Formation attributed to the MN3**
**biozone (Gagnaison et al., 2023). V. 2023/06 : The 6th International Chronostratigraphic Chart of the International Commission of**
**Stratigraphy (2023). GPTS : Geomagnetic Polarity Time Scale.**

## 2.3 Biostratigraphic age of the continental sands and nodule-bearing s1 bed

MN biozones were defined as a tool for inter-basin faunal comparisons (Mein, 1999). Limits of the zones are defined by 1) steps in mammalian evolutionary lineages (local evolution), 2) First Appearance Datum and/or Last Appearance Datum of species, 3) dispersal of taxa, and 4) faunal assemblages (Mein, 1999; Steininger, 1999). As discussed by Mein (1999), even when relatively inaccurate, the MN-zones are still a useful tool for regional correlation. For example, where local mammalian biozones are developed (e.g. the Mongolian Mammalian biostratigraphy proposed by Daxner-Höck et al., 2017), the MN system can still be employed since Europe and Asia often share taxa (Wang et al. 2013). However, we should keep in mind that correlation using the MN timescale is affected by ecological limits, latitudinal disparities, general diachronism in the dispersion of taxa, the presence of immigrant taxa (Mein 1999; Steininger, 1999) and local differences in taxa (even between neighbouring basins; Engesser and Mödden 1997).

Regionally, both the continental and marine Miocene sediments are known for their rich fauna of vertebrate fossils, including mammal taxa (Ginsburg, 2001). At Mauvières, vertebrate remains have been found within four horizons within the Orléanais Marls and Sands Formation (Figure 2a). The majority of fossils (>95%) are fresh and thus interpreted as syn-sedimentary and not reworked from older beds. In total, 53 taxa have been identified, most of them present in the s1 bed, the richest of the four fossiliferous horizons. The taxa are typical of the middle of the MN3 biozone (Gagnaison et al., 2023) (Figure 1b and Figure 2a).

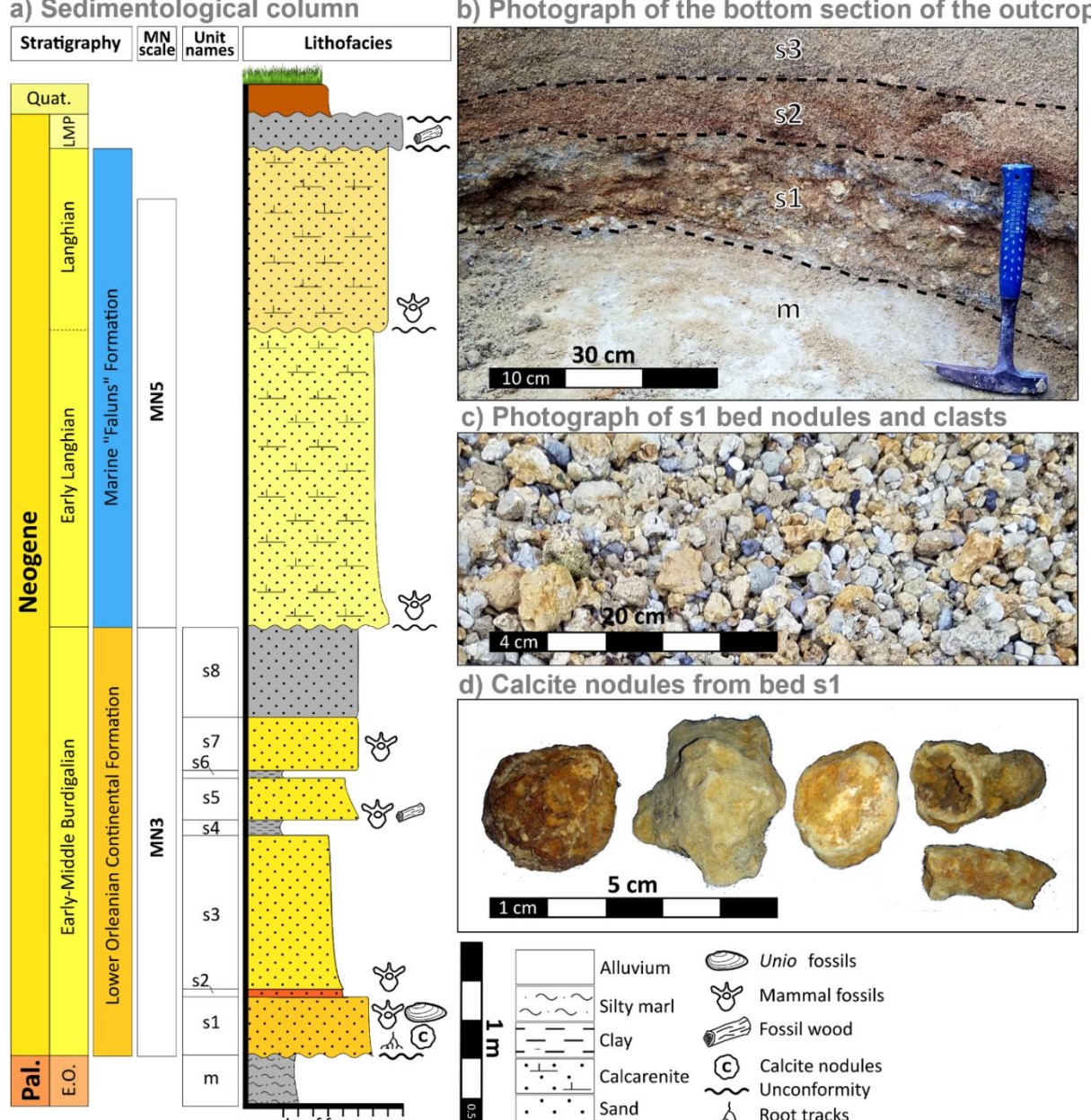

Figure 2: Geology of the Mauvières section. a) Sedimentological log and bed nomenclature (modified after Gagnaison et al., 2023). The calcite nodules are found in the Early-Middle Burdigalian basal sand s1. b) Photograph of the basal section of the outcrop, showing the basal Paleocene-Eocene silty marls unconformably overlain by an orange, very coarse fluviatile sand with mammal remains, freshwater mussel shells, root tracks and pedogenic calcite nodules. c) Granule and pebble fraction after sieving of the s1 sand. The fraction is dominated by pale-coloured calcite nodules. d) Photographs of representative calcite nodules from the s1 bed showing both spherical and cylindrical irregularly-shaped nodules of varying colour.

## 3 Materials and methods

### 3.1 Samples

Between 2020 and 2022, a series of geological sampling campaigns were undertaken at the Mauvières site. The sample material was sieved, washed, and dried. From the coarse separate (>2 mm), numerous nodules were collected and identified as vadose carbonate nodules (Gagnaison et al., 2023). The nodules are spherical to oblong, with a yellow-orange colour and a coarse aspect. Five of these nodules were selected for SEM elemental analysis and U-Pb dating (P00, P01, P02, P04, and P14), three nodules for powder X-ray diffraction analysis (XRD), and one nodule was prepared as a thin section for detailed microscopic analysis (MIOC4). For XRD analysis, each selected nodules were crushed in an agate mortar to create a fine powder. The five other nodules selected for U-Pb dating were sawn in half to reveal an internal surface. One half of each nodule was mounted in a 25 mm mould, mounted in epoxy resin, cured and polished, with the final polishing step employing 1 μm diamond suspension polishing fluid. The epoxy resin mounts were cleaned in an ultrasonic bath of deionized water for three minutes and imaged by optical microscopy. LA-ICP-MS U-Pb dating was undertaken on sample P00 to see if high quality age data can be obtained from the sample suite. Following LA-ICP-MS analysis of sample P00, it was repolished to remove the ablation rasters, cleaned in an ultrasonic bath of deionized water and carbon coated for SEM analysis. The other four samples were first carbon coated for SEM analysis, and later polished and then cleaned to remove the carbon before subsequent LA-ICP-MS analysis.

### 3.2 Optical microscopy

The resin pucks were imaged under reflected light using a Nikon Eclipse LV100 at the iCRAG Lab@TCD, Trinity College Dublin. Images were acquired at 5x magnification using a Nikon DS-Ri2 camera. Each tiled image is comprised of multiple frames stitched together by the Nikon NIS-Elements software. Each frame was taken with a square field of view of c. 2.8 mm in width and with an overlap of 10%. Transmitted and plane-polarised light images were also acquired for thin section MIOC4.

### 3.3 XRD

The powders were analysed using a Siemens/Bruker D5000 power X-ray diffractometer (Cu Kα radiation, 0.01 ° step−1 from 5 to 60 ° 2θ at 1 ° min-1, 4.5 hours per sample). Mineral identification was undertaken with DIFFRAC.EVA (Bruker) using the Powder Data File (PDF-4, The International Centre for Diffraction Data) (Gates-Rector and Blanton, 2019). XRD results and its interpretation are available in the Supplementary Materials.

### 3.4 SEM

The SEM analyses were carried out at the iCRAG Lab@TCD (Trinity College Dublin, Ireland) on a Tescan TIGER MIRA3 FEG-SEM equipped with a backscatter electron detector (BSE), two Oxford Instruments Ultim Max 170 mm$^2$ SSD EDX detectors and an X4 pulse processor. Scanning electron (SE) and BSE imaging and energy-dispersive X-ray spectroscopy

(EDS) analyses were acquired using an accelerating voltage of 20 kV and a working distance of 15 mm above the carbon-
coated pucks. The images and maps were processed using the AZtec version 6.1 X-ray microanalysis software suite (Oxford
Instruments).

## 3.5 Cathodoluminescence

Polished and uncovered carbon-coated thin sections for each sample were imaged using optical CL microscopy. CL images
were acquired at University College Dublin (UCD) using an HC5-LM hot-cathode CL microscope from Lumic Special
Microscopes, operated at 12.2 kV with a current density of 0.24 mA.mm$^{-2}$. No staining solution was applied prior to the
imaging.

## 3.6 LA-Q-ICP-MS

Laser ablation Q(quadrupole)-ICP-MS U-Pb dating was performed at the iCRAG Lab@TCD, Trinity College Dublin,
employing an Iridia 193 nm ArF excimer LA system (Teledyne Photon Machines, Bozeman, MT, USA) coupled to an Agilent
7900 Q-ICP-MS via 1.016 mm ID PEEK tubing and a medium pulse interface. One sample (P00) was dated using a mapping
approach and follows the U-Pb imaging technique described in Drost et al. (2018), while the remaining samples (P00-repeat,
P01, P02, P04, P14) were analysed by static spot analysis. For the latter, signal smoothing was achieved by inserting a mixing
chamber (Glass Expansion) between medium pulse interface and torch. Details on the specific analytical protocol and operating
conditions are given in the supplementary material (Supplementary Table 1-6 and Supplementary Document 1). This includes
the selection criteria, regions of interest, map dimensions and time-equivalents for all selected pixels and pixel groups ('pseudo-
analyses') for the sample analysed with the mapping approach, and the laser pit locations of the samples analysed by spot
ablation. Supplementary Tables are available on the Zenodo repository system (Monchal et al., 2024) while Supplementary
Document and Figure are available with the online version of this manuscript.
Samples were first screened for suitability using line scans. Samples and sample area yielding high initial Pb concentrations
and low μ throughout were omitted from further analysis. Similarly, samples areas with U ≤ 10 ppb were ignored as the young
sample age would result in very low concentrations of radiogenic Pb. Final locations for U-Pb analysis were selected according
to the results of the test line scans in combination with mineralogical and textural observations from optical microscopy and
from chemical information obtained by SEM-EDS mapping. In each dated nodule, we targeted calcite zones with minimal
incorporation of other phases. For the mapping experiment, this resulted in multiple groups of raster lines spread out across
the nodule surface. Final ROIs for data extraction were chosen to represent zones that may be interpreted as cogenetic and thus
a single age population constraining cementation. However, samples P01, P02, P04, and P14 did not feature large enough
coherent calcite areas with Pb/U ratios favourable for efficient and reproducible use of the mapping protocol. Spot analysis
was subsequently performed on those samples, using the chemical information from the SEM and LA-ICP-MS maps to help
site the spot analyses. Additionally, the U-Pb mapping data from sample P00 was also augmented by a static spot ablation
experiment.
The mapping session employed a laser spot size of 80 µm square, a repetition rate of 50 Hz and a fluence of 2.5 J/cm$^2$ while
moving the sample along successive linear rasters with 30 µm/s under the static laser beam. Samples were bracketed by NIST
SRM 614 glass as the primary standard, WC-1 calcite for matrix-matching the $^{206}$Pb/$^{238}$U ratio (Roberts et al., 2017) and Duff
Brown Tank limestone as quality control material (Hill et al., 2017). The total analysis time for sample P00 was c. 34 minutes.
Spot analysis employed 85 µm diameter spots, a repetition rate of 12 Hz, 480 shots (40s) and a fluence of 2.2 J/cm$^2$. Again,
NIST SRM 614 was used as the primary standard, but Duff Brown Tank limestone was used for matrix-matching of the
$^{206}$Pb/$^{238}$U ratio as it is closer in U concentration and age to the samples than WC-1. Gas settings (optimised daily), analyte
menus and integration times for all analytical sessions are reported in Supplementary Doc 1 along with the data processing
protocols used.
Uncertainties on dates in the text and figures are quoted at the 2σ or 95% confidence level, respectively. The geochronological
results are presented with two uncertainties; the first is an estimate of the session uncertainties, while the second is propagated
with full systematic uncertainties (e.g., the uncertainty on the reference age of WC-1 (maps) or DBT (spots) respectively, the
decay constant uncertainties, and the 2% long-term reproducibility of secondary age reference materials in the laboratory; see
Supplementary Document 1).

## 255 4 Results

### 256 4.1 Petrographic observations

The samples are composed of transparent, mostly rounded quartz grains with some more angular crystals, set in a pale orange-
yellow cement with vein-like cavities, partially filled with carbonate crystals (Figure 3). The majority of the samples exhibit a
main cavity that in some cases branches out *via* micro-cracks, typical of alpha type paleosoil (Wright, 1990). We can
distinguish two stages of formation. The first stage involves the formation of sedimentary concretions around roots. The
concretions are rich in quartz and cemented by clear carbonate as observed under the optical transmitted light microscope
(Figure 3). After decomposition of the roots, sparry carbonate crystals precipitated predominantly into free space producing a
brown layer on the edge of the cavity and filling the micro-cracks. The host-rock is composed of touching or floating
terrigenous clastic elements such as quartz in a clotted carbonated matrix with authigenic goethite. The host-rock is also cross-
cut by rhizolith root tubules, traces of which are still visible (Figure 3A). These relics of paleo-roots are expressed by the stack
of several layers of dark microbial micrite linings (Figure 3A) and some holocrystalline microsparite. The presence of
holocrystals is dependent on the degree of microbial activity and the root structure (i.e. main axis *vs* lateral roots). These early
pedogenic carbonate crystals (e.g. the calcite crystals in Figure 3B-C) are classically found in many paleosols (e.g. Wright,
1987; Esteban and Klappa, 1983; Bain and Foos, 1993; Alonso-Zarza, 2003).
Sample MIOC4 is a representative nodule from the s1 bed that exhibits evidence of primarily calcified root traces (Figure 3;
see also Gagnaison et al., 2023). No evidence of later crystallisation nor recrystallisation was detected, with the calcite spar
homogeneous and unzoned (Figure 3B-C). Moreover, micro-cracks and alveolar structures are commonly found without
calcite crystallisation (Figure 3A), especially where the primary root was located. When calcite crystals are present, they are
typically associated with lateral roots.

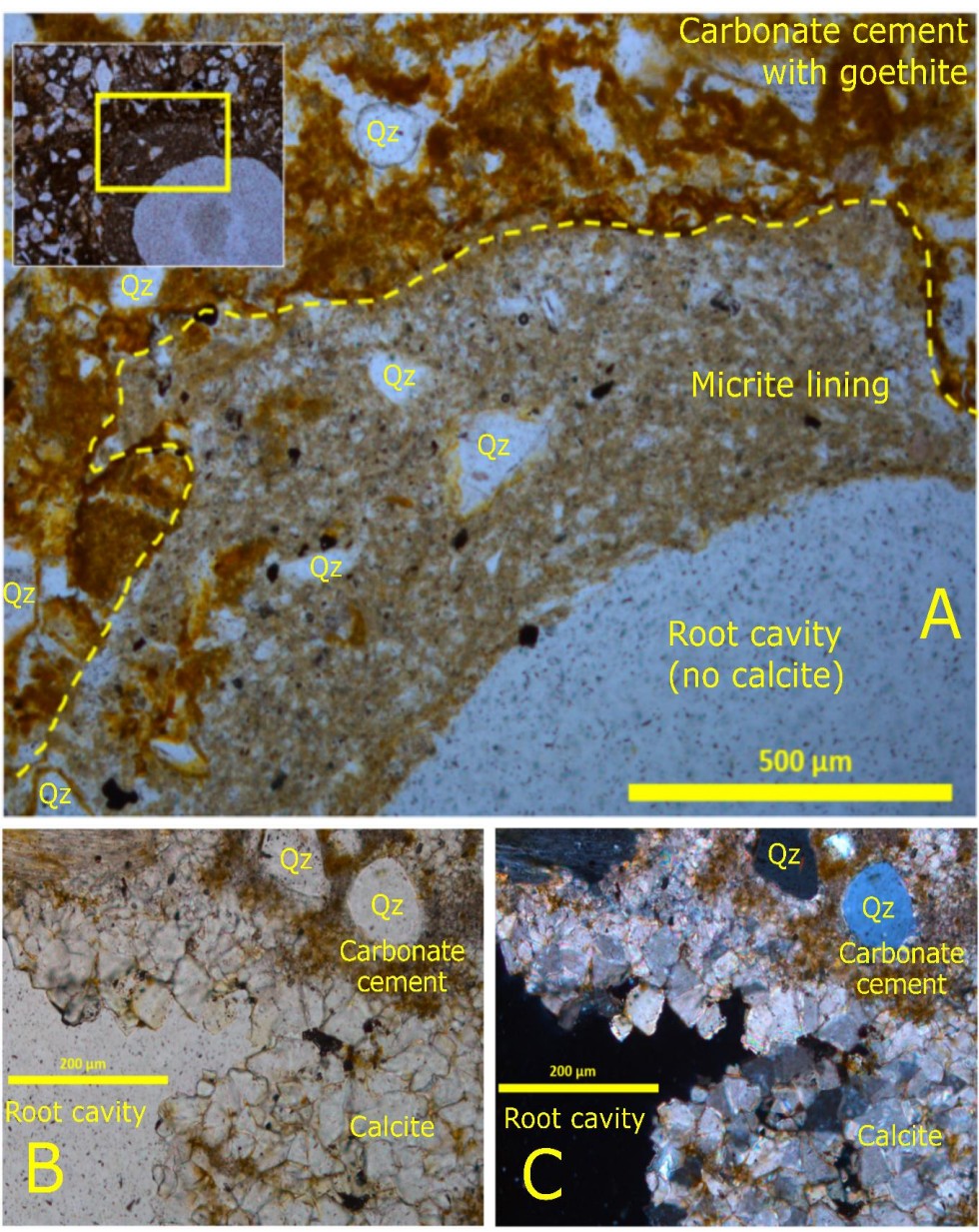

**Figure 3: Optical microscope photography of sample MIOC4. A) Primary root structure with a dark microbial micrite lining –**
**dashed yellow line highlights the boundary of the external part of the microbial micrite lining. An alveolar structure can be seen on**
**the zoomed out insert at the top of the microphotograph (PPL). B) Sparry calcite crystals; a lateral root perforation is on the lower**
**left side of the microphotograph (PPL) and C) (XPL). Q = Quartz.**

## 4.2 SEM-EDS elemental mapping

The SEM-EDS maps of the five dated nodules reveal that the nodules are composed of poorly sorted angular Si-rich minerals cemented by a Ca-rich phase (Figure 4). The two phases are interpreted respectively as quartz and calcite based on optical microscopy and the PXRD results. The cemented sand also contains grains rich in Si and K, Na interpreted as feldspar and in agreement with the results of PXRD. Large cavities, mostly branching or rounded are present in all the nodules. These cavities are lined by a pure Ca-rich phase interpreted as calcite that precipitated into the free cavity space. In some locations, quartz-free calcite crystals have filled the cavities entirely. These zones of pure calcite were subsequently targeted for LA-ICP-MS U-Pb dating.

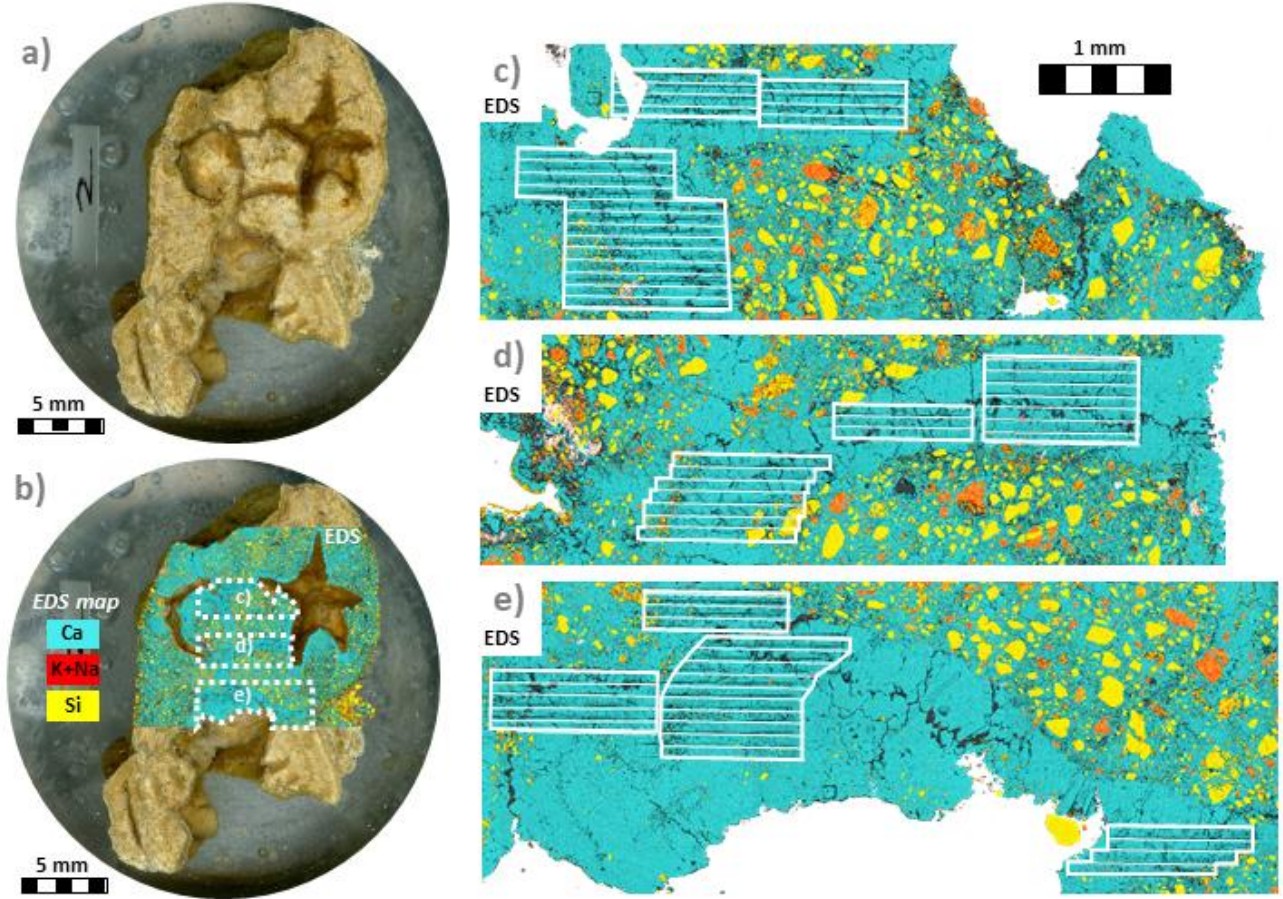

**Figure 4: Photographic montage of nodule P00 in a polished resin puck. a) optical microscopy image b) the same image overlain by a partial EDS map of the nodule showing Ca (a proxy for calcite, blue), Si (a proxy for quartz, yellow), and K+Na (a proxy for feldspar, red). The location of the EDS maps in c), d), and e) are represented by the dashed white polygons. c), d), and e) EDS maps showing the LA-ICP-MS ablation zones and line scans for the P00 nodule. Pure calcite veins were targeted, avoiding the zones of calcite-cemented quartz-rich sand. See Supplementary Materials for pictures and EDS map of the other samples (Supplementary Figure 1).**

**4.3 Cathodoluminescence imaging**
The calcite-cemented sands in the concretions show a complex pattern of dull brown and orange to bright yellow luminescent
calcite cementing quartz and minor feldspars which are highly (and variably) luminescent. Sparry carbonate crystals infilling
cavities and fractures show strong oscillatory CL zoning at the <10 µm scale (Figure 5). The calcite growth in the fractures
oscillates between non-luminescent, dull brown to orange luminescence, and bright yellow and orange luminescence. Growth
morphologies from CL are euhedral to occasionally subhedral blocky with no recrystallisation of the oscillatory zoning
observed.

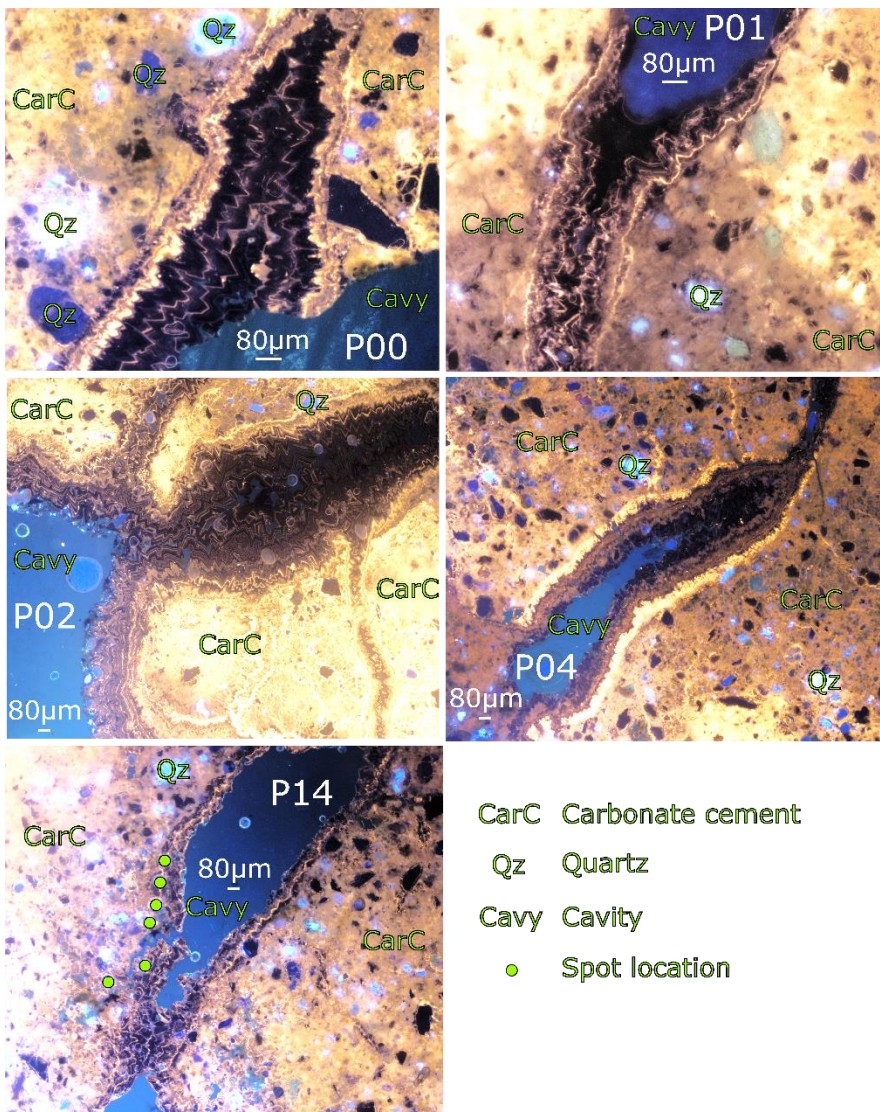

**Figure 5: Cathodoluminescence images from the samples at 5x (P02/P04/P14) or 10x (P00/P01) magnification illustrating the**
**oscillation between non-luminescent dull brown and orange luminescent zonation in the calcite crystals. Spot locations are shown**
**on the P14 photo showing that the outer margins of the calcite zones were ablated.**

### 4.4 LA-ICP-MS U-Pb dating

Calcite crystals that have precipitated freely inside the cavities were targeted for geochronology analysis (Figure 6) as they are believed to have precipitated rapidly after the formation of the paleosol (see section 2.2 and Rasbury et al., 1997). The mapped areas in P00 targeted zones of pure calcite based on the SEM-EDS mapping. A Ca filter (e.g. retaining pixels with Ca > 350 000 ppm) was applied on the P00 map to exclude any inclusions, cracks, epoxy resin or the host sedimentary rock and this filter removed c. 7% of the pixels from the maps. The average U content is ~ 10 µg/g (ranging from 9 to 13 µg/g), while the average Th content is ~ 0.7 µg/g (ranging from <0.1 to 2.5 µg/g; see Supplementary Table 1) resulting in Th/U ratios of <0.01 to <0.2. Significant initial Pb concentrations (~0.44 to 33 µg/g) and the long half-life of Th in combination with the young age of the samples render the radiogenic ingrowth of radiogenic $^{208}Pb$ negligible ($^{208}Pb_{common}/^{208}Pb_{radiogenic}$ ~2800 to 12000). Therefore, we used the empirical cumulative distribution function of the $^{238}U/^{208}Pb$ channel for pooling of the filtered pixel data into pseudo-analyses. The $^{238}U/^{208}Pb$ channel is a good estimate of the µ ratio between parent U ($^{238}U$) vs initial Pb ($^{204}Pb$) as the total $^{208}Pb$ concentration is a robust proxy for the initial $Pb_{common}$ component.

The spot U-Pb data were corrected post-analysis for any ablation that went through the calcite. This correction employ the visual inspection of peaks for a significant change in Ca, Pb, Th or U composition that indicate a change in the phase ablated. The U-Pb spot analyses on samples P01, P02, P04 and P14 yielded dates of 18.8±2.7/2.7 Ma, 19.11±0.84/0.94 Ma, 19.0 ±2.3/2.3 Ma, 19.4±2.7/2.7 Ma, respectively, while sample P00 yielded dates of 19.3±1.3/1.4 Ma (mapping) and 19.7±1.5/1.6 Ma (spots) (Figure 7). A radial plot and weighted average age were calculated using the five dates from spot analysis and their respective internal uncertainties (session estimates) featuring a Mean Square Weighted Deviation (MSWD) and chi-square ($p[\chi^2]$) test representing how good the results are fitting to the statistic value. The full systematic uncertainties (section 3.6) were propagated onto the resultant age (radial plot or weighed average) calculation. The radial plot in Figure 8 shows a single age group at 19.22±0.66/0.79 Ma ($p[\chi^2] = 0.96$) and a weighted average age was calculated at 19.21±0.64/0.77 Ma (MSWD=0.16; $p[\chi^2] = 0.96$; see Figure 8). All U-Pb spot data were also plotted in the same Terra-Wasserburg space with their individual propagated uncertainties providing a result of 19.1±0.56/0.71 Ma. The radial plot single group age of 19.22±0.66/0.79 Ma is the preferred age adopted in this study as discussed in section 5.2.

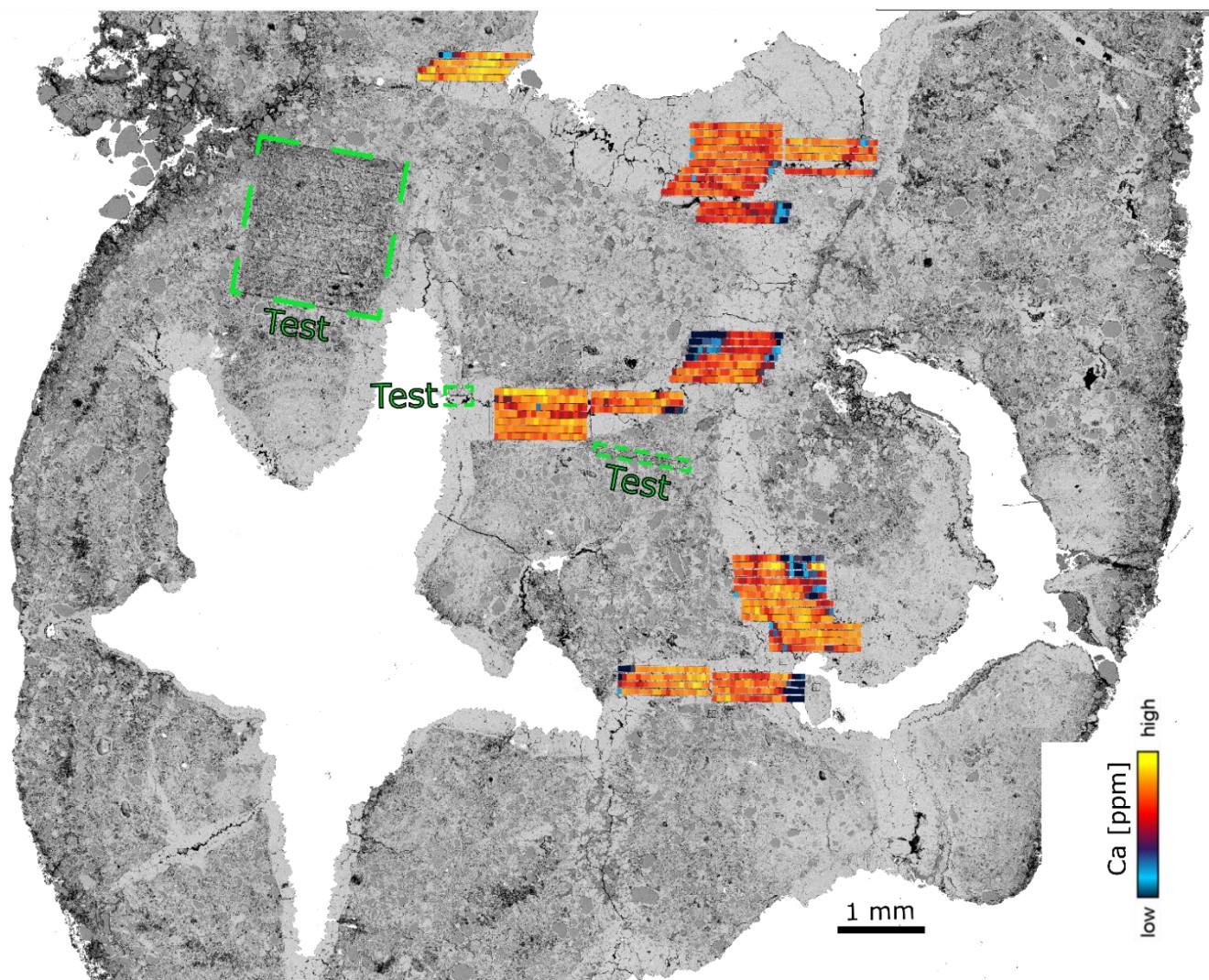

**Figure 6: BSE image of P00 overlain by the LA-ICP-MS line rasters used to extract the age. This figure shows that the pristine**
**calcite was targeted by our analysis.**

## 5 Discussion

### 5.1 Accuracy and precision of the U-Pb ages

The imaging techniques (optical microscopy, SEM-EDS and LA-ICP-MS mapping) have differentiated zones of pristine
calcite and the pervasive cementation of the nodules. Optical microscopy evidence favours the hypothesis of preservation of
pristine calcite in our samples (see Results section 4.1). In addition, prior to their extraction from the s1 bed, all the pedogenic
nodules (along with clasts and fossil material) were coated with an impermeable clay layer, which likely hindered subsequent
passage of fluid into the nodules. The clay coating is interpreted as syn-sedimentary (see figures 2b-d and Gagnaison et al.,
2023). This sealed system is another argument in favour for the preservation of pristine calcite (Perry and Taylor, 2006) in the
nodule interiors (See Figures 4 and 5). The nodule morphology is preserved (not rolled or broken) and does not feature any
sign of compaction nor internal collapse which supports the hypothesis of non-reworked nodules. Tubular nodules have also
been found perpendicular to the stratigraphy, thus clearly marking the former position of the root. The Eocene-Oligocene marls
(m on Figure 2) below the s1 bed do not contain nodules, further supporting the hypothesis that the nodules found in the s1
bed are in-situ.
The growth morphologies from optical and CL microscopy indicate gradual growth competition took place, indicative of a
crystallisation in a cavity that remained open (e.g. Wendler et al., 2016; Prajapati et al., 2018). The oscillatory zoning with
multiple bright concentric subzones observed under CL (Fe is the main CL quencher and Mn the main activator) can be
explained by small yet rapid variations in Eh/pH conditions accompanied by changes in oxidation state (e.g., Pagel et al.,
2000). With increasing oxidation, $Fe^{2+}$ and $Mn^{2+}$ sensitized by $Pb^{2+}$ and/or $Ce^{3+}$ (Pagel et al, 2000) are replaced by $Fe^{3+}$ and
$Mn^{3+}$ or $Mn^{4+}$ ions (e.g. Richter et al., 2003; Boggs and Krinsley, 2006). A plot of log [Fe]/log [Mn] ppm can predict if calcite
will be bright, dull or non-luminescent in CL (Machel and Burton, 1991; Boggs and Krinsley, 2006) (see Supplementary
Material). The specific CL patterns are consistent with redox fluctuations caused by water table fluctuations in a vadose or in
a fluid-saturated environment (Mason, 1987; Barnaby and Rimstidt, 1989), which is also in agreement with the previous paleo-
environment reconstitutions for the s1 bed (Gagnaison et al., 2023). Given the above observations and since the oscillatory
zoning is continuous, we interpret a single continuous event of calcite formation to have occurred inside the nodules. The
differing thickness of the CL bands appears related to the size of each cavity in the nodules, with P00/P01/P02 having the
largest cavities and bands while nodules P04 and P14 have thinner bands.

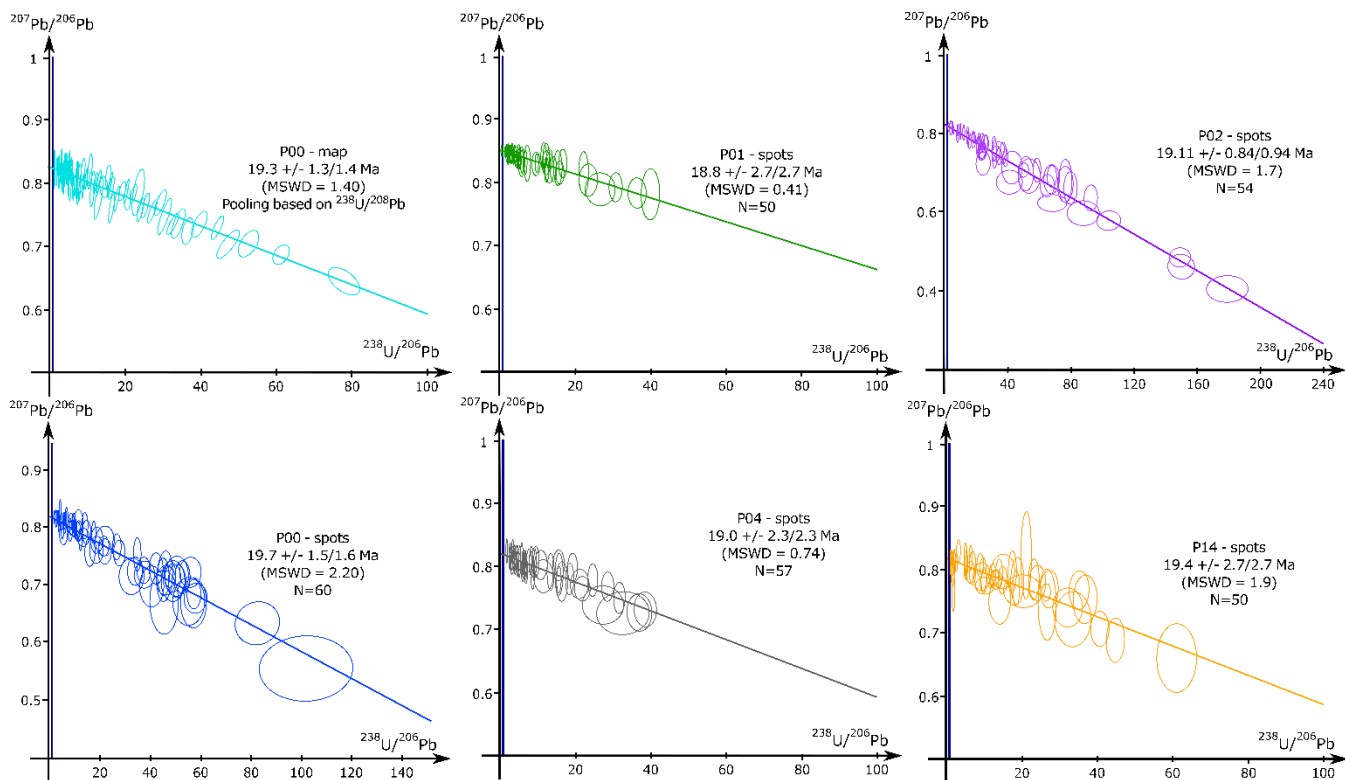

**Figure 7: Tera-Wasserburg concordia diagrams and lower intercept ages of all samples. For the map analysis of P00, the pooling was based on the ECDF $^{238}$U/$^{208}$Pb. For the spot analysis, the number of spots is indicated by N.**

The LA-ICP-MS mapping technique adopted herein is recognised for its potential (see Rasbury et al., 2023) in dating pedogenic nodules by allowing the selection of only pristine calcite in the extraction and processing of the U-Pb data. However, only one sample had large enough coherent zones of pristine calcite with Pb/U ratios suitable for U-Pb dating and a spot analysis strategy was used to date the remaining four samples. All six results yield ages with a precision of 5 to 14%, which is considered precise for LA-ICP-MS carbonate U-Pb dating of such young samples (Roberts et al., 2020). The accuracy of our data set can be assessed by the fact that the five samples provide the same age and initial $^{207}$Pb/$^{206}$Pb within uncertainties, along with the radial plot confirming that there is only one age group (Figure 8). The accuracy of the mapping experiment is also demonstrated by the similar dates (within uncertainties) yielded using three different isochron approaches. The mapping approach data for sample P00 (using $^{238}$U/$^{208}$Pb as the pooling channel) yields 19.3±1.3/1.4 Ma for the TW intercept age, 19.6±1.7/1.8 Ma for the $^{238}$U/$^{208}$Pb$_{common}$ isochron (e.g., Getty et al., 2001) and an 86TW age (Parrish et al., 2018) of 19.4±1.6/1.7 Ma (section 3.4 and Supplementary Table 1).

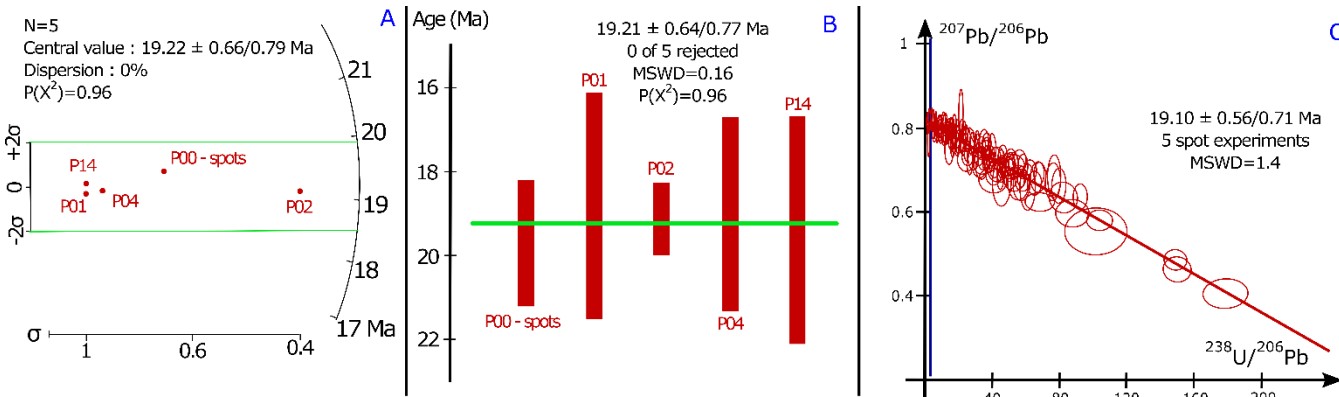

Figure 8: A) Radial plot and B) weighted average of the samples used in this study. Radial plot central value and the weighted average value are indicated with 2σ internal uncertainties (session estimates). The full systematic uncertainties (section 3.6) were propagated onto the resultant age calculations with the same method as for the individual sample ages. C) Tera-Wasserburg concordia diagram of all five spot ablation experiments. See text for the interpretation and discussion of the data.

**5.2 Age of the nodules and paleosol**

Our age data are compatible within uncertainty with the proposed biostratigraphic age of the continental biozone MN3, which is correlated with the Burdigalian marine stratigraphic age (20.44 - 15.98 Ma; Cohen et al., 2013 [updated 2023/09]) and the Orleanian continental stratigraphic age (19.5 - 14.2 Ma; Hilgen et al., 2012). Dating the pedogenic calcite should provide a minimum age for the paleosol formation (Rasbury et al., 1997). The nodules are found within the same sedimentary layer (section 2.2) and we can therefore reasonably assume that the crystallisation of the calcite inside each nodule arose from the same process(es). Even if these process(es) involve multiple phases of growth, we do not see any evidence of incremental growth of more than one generation of calcite from petrography, CL and SEM-EDS mapping. The U/Pb dates obtained on these five nodules are identical within age uncertainty of our method (Fig. 8) and do not exhibit evidence for more than one stage of calcite growth, diachronous growth across different nodules or a substantial time span between initiation and termination of calcite formation. We therefore assume that formation of the analysed nodules (which are identical within age uncertainty of our method) was effectively synchronous.

To determine the minimum age of nodule formation, there are several possible approaches: 1) the U/Pb date with the lowest uncertainty (P02: 19.11±0.84/0.94 Ma), 2) the date derived from the combined TW regression of spot analyses from all five samples (Fig. 8C; 19.10±0.56/0.71 Ma), 3) a weighted mean of the U/Pb dates from all analysed samples (Fig. 8B; 19.21±0.64/0.77 Ma) or 4) the radial plot age (Fig. 8A; 19.22±0.66/0.79 Ma). The former two methods may introduce some bias as they may overly rely on the data points with the highest $^{238}U/^{206}Pb$ ratios coming all from the same sample (P02), while the latter two methods put more emphasis on the similarity of the results associated with individual samples. The radial plot shows only one age group, and the central age from the radial plot and the weighted mean of the TW intercept ages are identical within age uncertainty. The weighted mean age calculation assumes the data follows a normal distribution, while the radial plot assumes that the log of the values follows a normal distribution curve (Vermeesch, 2018). Geochronological data are less

likely than other data to conform to a normal distribution due to the presence of outliers and the range of age values must be
positive, thus the distribution is asymmetric (Vermeesch, 2018). The log of the outliers used in radial plots will smooth these
deviations and heteroscedastic variation (unequal uncertainties) and make it fit to the normal distribution curve (Galbraith et
al. 1999), which is why the radial plot central age is preferred. This age of 19.22±0.66/0.79 Ma for the s1 bed allows precise
correlation with other dated sequences, independently of the lithofacies or fossil assemblages present. This age is the first
absolute age for the continental Miocene facies of the Paris Basin and to the best of our knowledge the youngest U-Pb age
from pedogenic carbonates in the literature (Table 1).

**5.3 Biostratigraphic significance**

The MN (Mammal Neogene) stratigraphic timescale is based on faunal calibration. The appearance and disappearance of taxa
result in a given combination of species that can be linked to a given time (Mein, 1999). The MN scale incorporates a
stratigraphic component as well as classical stratigraphic correlations and magnetostratigraphy to help refine the age control
(Hilgen et al., 2012). MN units were initially defined without boundaries or clearly defined limits (e.g. Mein, 1975), but
nowadays the scale is often presented alongside a chronostratigraphic scale, with an absolute age associated with each biozone
boundary (e.g. Agustí et al., 2001; Van Dam et al., 2001; Aguilar et al., 2003; Gagnaison et al., 2023). The absolute ages of
the boundaries remain debated (see the example of MN3 below) due to diachronicity and incomplete paleontological and
magnetostratigraphical data (Fortelius et al., 2014; Ezquerro et al., 2022). Each zone is characterised by a specific fauna found
at a reference locality (for Europe these are mainly in Spain, France, and Germany) that can be asynchronous by up to 1 - 2
Myr in the Late Miocene (Van der Meulen et al., 2012; Fortelius et al., 2014; Ezquerro et al., 2022). The majority of the MN
zones have uncertainties attached to their age boundaries (Figure 9), while the application of the MN timescale typically
involves comparison to the most proximal and well-constrained reference section to circumvent potential diachronicity. Local
modifications to the MN timescale are thus often adopted for selected biozones (Hilgen et al., 2012; Van der Meulen et al.,
2012; Fortelius et al., 2014; Ezquerro et al., 2022).
To improve the precision of this scale, the incorporation of magnetostratigraphy has helped to better define the MN unit
boundaries within basins (e.g. Agusti et al. 2001; Kälin and Kempf 2009). Steininger (1999) used magnetostratigraphic data
to propose that magnetochron C6r (20.5 Ma) represented the base and C5Dr the top (18.5 Ma) of MN3. The top boundary of
MN3 was then extended to chron C5Cn.2r, dated between 16.6 and 17.2 Ma, based on magnetostratigraphy of sections in the
North Alpine foreland (Agusti et al., 2001). The MN3 faunal reference site was defined as Wintershof-West with a sedimentary
succession dated between 17.5 and 18.5 Ma (Hilgen et al., 2012) thus only partially covering the time interval defined by the
magnetochron ages. The MN3 boundaries were refined by magnetochron ages for C6r (19.979 Ma) and C5Dr (18.007 – 17.634
Ma) Ma, which are the chrons defined by Steininger (1999) as bracketing the MN3 biozone, the ages of these magnetochrons
are subsequently updated by Raffi et al. (2020). It should be noted that magnetostratigraphy requires thick sections (typically
>10 m thick profiles) and cannot always be employed. Our results are compatible within uncertainties with the different
magnetochron ages proposed for MN3 and are not challenging the actual consensus around the absolute age of the base or the
top of MN3 (Figure 9).
Absolute U-Pb dating of in-situ pedogenic carbonates enables a better understanding of the spatio-temporal distribution and
evolution of continental mammalian faunas. This method is not affected by the limits detailed above (i.e. diachronicity, index
fossil scarcity, insufficient profile thickness), thus offering a reliable opportunity to improve the local constraints on the MN
scale. Our age is compatible with an early Orleanian stage assignation (Figure 1) and the MN3 unit (Hilgen et al. 2012). The
age constraints on the Mauvières fossil locality are thus significantly improved by our results, but it should be noted that an
age for one locality does not improve the precision of the MN3 boundaries at a European scale. Therefore, more studies
employing similar method are needed for further improvement of the MN scale, especially zones with large uncertainties such
as MN3.

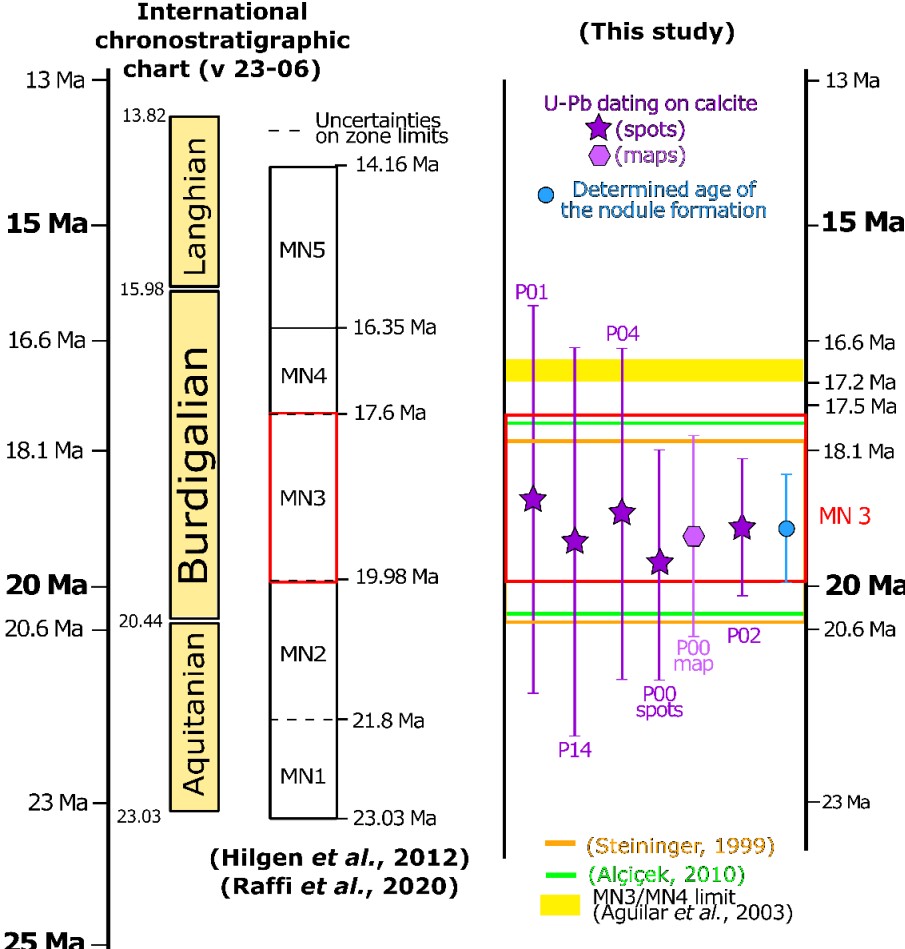

**Figure 9: Overview of MN timescales in the literature compared to the age data from this study. The red box defining the currently**
**accepted boundaries of the MN3 biozone is taken from Raffi et al. (2020) by taking the base of magnetochron C6r at 19.979 Ma and**
**the top of C5Dr as 17.634 Ma. The age of nodule formation is the result of the radial plot using the six U-Pb geochronology dates**
**and their respective internal uncertainties; the full systematic uncertainty was propagated on to the radial plot result age calculation**
**(see sections 4.4 and 5.2).**

## 6 Conclusions

The application of LA-ICP-MS U-Pb dating of carbonate pedogenic nodules as employed in this study is a robust and reliable way to provide absolute age data for terrestrial strata. Our samples yield a precise and accurate age of 19.22 ± 0.79 Ma in accordance with earlier biostratigraphic estimates (Orleanian), demonstrating the suitability of the method and confirming the feasibility of the technique to dating continental sedimentary facies that do not contain any index fossils or volcanic horizons such as lavas or ash beds.

Our results are in good agreement with the biostratigraphic age (MN3 of the Neogene Mammalian timescale) of sedimentary horizon s1 from Mauvières (Gagnaison et al., 2023) and represent the first absolute age constraint for the MN3 unit in France. This absolute age dating approach has the potential to advance chronostratigraphy and climatic reconstructions (Liivamägi et al., 2021) by improving inter-basin correlations in continental successions and extending such correlations to the marine sedimentary record. In order to refine the geochronological constraints, the use of a more precise reference material would decrease the external uncertainties (i.e. ASH15, Nuriel et al., 2021; JT, Guillong et al., 2020;RA138, Guillong et al., 2024). The protocol for U-Pb dating of carbonate nodules proposed by Aguirre Palafox et al. (2024) offers a uniform approach and a basis for comparisons between studies. While their study was published during the review process of this manuscript, it should be noted that our study nevertheless broadly conforms with this protocol.

## Author contribution

VM contributed to the conceptualisation, the formal acquisition, the investigation, the methodology, the project administration, the visualisation and the writing (initial draft and edits). RR contributed to the conceptualisation, the formal acquisition, the investigation, the methodology, the project administration, the visualisation and the writing (edits and part of initial draft). KD contributed to the methodology, the supervision and the writing (edits). CG and BM contributed to the resources, the conceptualisation and the writing (edits). RT contributed to the writing (edits). DC contributed to the supervision, the funding acquisition and the writing (edits).

## Competing interests

The authors declare that they have no conflict of interest.

## Acknowledgments

The authors would like to acknowledge the Poirier family who allowed us to sample the shell bed (falun) in Mauvières quarry. We would also like to thank Didier Memeteau and Bruno Cossard for assistance sampling the nodules. We acknowledge the support of Science Foundation Ireland, the Environmental Protection Agency, and Geological Survey Ireland under

Investigators Programme grant 15/IA/3024. Comments by reviewers Perach Nuriel, Andreas Möller and associate editor Axel
Schmitt signficantly improved this manuscript and are gratefully acknowledged.

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
