# Peer review of "U-Pb dating on calcite paleosol nodules: first absolute age constraints on the Miocene continental succession of the Paris Basin"

_EGUsphere, 2024_

## Referee Comment (RC1)

Review notes for Monchal et al paper entitled "U-Pb direct dating on calcite paleosol nodules: first absolute age constraints on the Miocene continental succession of the Paris Basin "

This paper demonstrates the great applicability potential of latest advances in laser ablation ICPMS U-Pb dating of carbonate material. Paleosol nodules are some of the more complicated materials both in terms of their microstructures and history of diagenesis, thus a successful attempt to date these sedimentary structures has important Implications for many studies to follow and is well deserve for publication in *Geochronology*.

My comments will be focused on the dating part and less on the implications of the dated material for the reconstruction of the Miocene continental succession in the Paris Basin. I am sure other reviewers can cover that part of the paper.

My first comments relate to the study of these texturally complex structures. As well demonstrated, imaging and geochemical understanding of these structures is essential for dating them. The state-of-the-art mapping technique that was developed by the team at Trinity College Dublin is a good application in this type of samples, however, I wonder if some more basic cathodoluminescence imaging could be beneficial here? From my experience, these samples have a long diagenetic history that is hard to observe by SEM/XRD/BS imaging. For example, consider the below image, you can see a clear complexity of fluid composition and/or precipitation conditions as well as overprint phase (bright luminescence). Perhaps, if not too late, the author could provide CL images of the studied sample? I think this could be very nice addition to the textural study if possible.

[Figure]

[Figure]

CL image of carbonate nodules from the Jura Mt. in France (unpublished data).

My second comments relate to the obtained U-Pb ages and interpretation. The authors present 6 TW plots (figure 7), however, at the end the whole temporal constraint is based on statistical approach (radial plot in figure 8). I wonder if this interpretation misses a bit the potential diagenetic history of the sedimentary section (from 20.4 and 18 Ma)? One possibility is that the oldest age will better constrain the deposition age and younger ages correspond to diagenetic processes. Alternatively, the oldest ages correspond to reworked material and the youngest ages are the depositional time. I don't know the right answer, but I think there is room for discussion on these options. It will be also interesting to look at sedimentary rates with the obtained ages if available? to see if age of 20 or 19 Ma makes a difference? I don't know how important this 1 Ma difference is, but it is something to be discussed maybe in the text?

In addition, what is the TW age for all spot analyses (of all 6 samples)? And what is the MSWD? If all nodules are considered as the same age and we do not attempt to study their diagenetic history, then why not presenting this age as the age for the section? I would have plotted it myself, but I could not find the "spot" analyses data (to be included?).

Considering the overlap uncertainties, this will not make much difference, however, if in the future the authors could use better primary standards (minimal uncertainties of about 2.5% for the WC1) such as JT, RA138 or ASH15 then the diagenetic history of such samples could be potentially resolved.

More general comments:

Introduction
I think a paragraph explaining the formation and diagenetic history of carbonate nodules could be important here. There are lots of studies using mineralogical and geochemistry characterization of such nodules. It will help to associate their precipitation time with sedimentation process, including key observations for their association.

Line 60 – You could add, *Microcodium* dating from Paleocene–Eocene Thermal Maximum in the Southern Pyrenees (Spain) - Prieur et al 2024 , *Geology*.

Line 125 - I am not convinced by the assertion that these nodules are not reworked, it will help to provide more microstructural evidence, such as preservation of nodules morphology, age comparison in different sections etc. If not possible to provide, perhaps it is better to mention the possibility and what different it will make? (if reworked from S1-S3? Age is slightly older?).

Section 3.2. Will be great to add cathodoluminescence imaging section if possible.

Line 215 onward – I think the main limitation for the precision here is the primary reference material and not the secondary, as might be understood from this paragraph. The mapping technique is a real state-of-the-art approach; however, it is limited here by the used primary reference material (RMs - WC1), with a minimal uncertainty of 2-2.5%, thus, ages will be plus minus 0.5 Ma for an age of 20 Ma. This notion may help to push forward the use of better RMs in the future.

Line 235 – how do you evaluate recrystallization without XPL imaging on thin-sections and/or cathodoluminescence? Crystal growth morphology is not well documented, so this is not entirely supported by the data provided. For example, Figure 3b and c may look like a single cementation phase with PPL but could show very different CL and XPL characteristics. I don't mean to be difficult here, but I have seen many examples that show this pattern.

Line 320 – The problem with multiple phases of growth is that they can occure at different stages of the daiagenetic history. Also it is very hard to observe incremental growth in the provided images.

Line 325 – I find the statistical explnation unneseserly complicated, I think it can be simplified by first provideing comparison for the different approaches such as age constraint base on the:

1.  age with lowest uncertenties: 19.11 ±0.94 Ma
2.  TW plots of all "spot" analyses : ???
3.  radial plot: 19.34 ±0.73 Ma
4.  mean weighted average : 19.32 ±0.73 Ma
5.  oldest age, to account for post formation daiagenetic processes: 20.4 ±1.6 Ma
6.  Youngest age, to account for potential reworked contribution: 18 ±3.2 Ma

Then you can explain why you choose option number 3?

Overall, I enjoyed reading the paper and I think it is well written and should be published in geochronology. I think a bit more textural control (CL imaging) and discussion on potential interpretation of the data could improve the paper and make it an important contribution for future studies to come that will take similar approaches.

With best wishes,
Perach Nuriel

---

## Author Response (AR2)

Response to reviewer: Perach Nuriel – reviewer comments are in blue italics.

We would like to thank the reviewer for her constructive comments that have improved the paper.

*My first comments relate to the study of these texturally complex structures. As well demonstrated, imaging and geochemical understanding of these structures is essential for dating them. The state-of-the-art mapping technique that was developed by the team at Trinity College Dublin is a good application in this type of samples, however, I wonder if some more basic cathodoluminescence imaging could be beneficial here? From my experience, these samples have a long diagenetic history that is hard to observe by SEM/XRD/BS imaging. For example, consider the attached image, you can see a clear complexity of fluid composition and/or precipitation conditions as well as overprint phase (bright luminescence). Perhaps, if not too late, the author could provide CL images of the studied sample? I think this could be very nice addition to the textural study if possible.*

CL imaging was envisaged at an early stage of drafting of this paper, however the CL in our home institution is not optimized for carbonates as it is a hot cathode setup on a SEM (a single image takes hours and the end result is poor). A new collaboration allowed us to access an optical-based cold cathode CL system better suited for carbonates. We incorporated the CL images and relevant discussion to this paper which provides more context on the sample petrography and the resultant interpretation.

*My second comments relate to the obtained U-Pb ages and interpretation. The authors present 6 TW plots (figure 7), however, at the end the whole temporal constraint is based on statistical approach (radial plot in figure 8). I wonder if this interpretation misses a bit the potential diagenetic history of the sedimentary section (from 20.4 and 18 Ma)? One possibility is that the oldest age will better constrain the deposition age and younger ages correspond to diagenetic processes. Alternatively, the oldest ages correspond to reworked material and the youngest ages are the depositional time. I don't know the right answer, but I think there is room for discussion on these options. It will be also interesting to look at sedimentary rates with the obtained ages if available? to see if age of 20 or 19 Ma makes a difference? I don't know how important this 1 Ma difference is, but it is something to be discussed maybe in the text?*

We have added a new paragraph to the discussion section that considers these points.

*In addition, what is the TW age for all spot analyses (of all 6 samples)? And what is the MSWD? If all nodules are considered as the same age and we do not attempt to study their diagenetic history, then why not presenting this age as the age for the section? I would have plotted it myself, but I could not find the "spot" analyses data (to be included?).*

The analytical data is available at the following updated zenodo link :
https://doi.org/10.5281/zenodo.14500416
According to the author instructions for *Geochronology,* the dataset was referenced to in the text using the hyperlink provided above. A TW plot using all the analytical data from the spot analyses of each sample was added to the manuscript and discussed.

*Considering the overlap uncertainties, this will not make much difference, however, if in the future the authors could use better primary standards (minimal uncertainties of about 2.5% for the WC1) such as JT, RA138 or ASH15 then the diagenetic history of such samples could be potentially resolved.*

This is one of the current limitations of the method, we add this point to a future work paragraph at the end of the conclusions.

*Introduction*

*I think a paragraph explaining the formation and diagenetic history of carbonate nodules could be important here. There are lots of studies using mineralogical and geochemistry characterization of such nodules. It will help to associate their precipitation time with sedimentation process, including key observations for their association.*

We added a new paragraph to the introduction developing this point, and it is also useful for the discussion on the CL imaging that follows later.

*Line 60 – You could add, Microcodium dating from Paleocene–Eocene Thermal Maximum in the Southern Pyrenees (Spain) - Prieur et al 2024 ,Geology.*

This point and its associated reference was added to the text.

*Line 125 - I am not convinced by the assertion that these nodules are not reworked, it will help to provide more microstructural evidence, such as preservation of nodules morphology, age comparison in different sections etc. If not possible to provide, perhaps it is better to mention the possibility and what different it will make? (if reworked from S1-S3? Age is slightly older?).*

The nodule morphology is preserved (not rolled or broken) and does not feature any sign of compaction nor internal collapse which supports the hypothesis of non-reworked nodules. Tubular nodules have also been found perpendicular to the stratigraphy, thus clearly marking the former position of the root. The Eocene marls (m on Fig. 2) below the s1 bed do not contain nodules, further supporting the hypothesis that the nodules found in the s1 bed are in-situ. Furthermore, the new CL imaging indicates only one growth phase for the sparry calcite crystals, which indicates no major post-formational proceses affected the nodules.

*Section 3.2. Will be great to add cathodoluminescence imaging section if possible.*

*Line 235 – how do you evaluate recrystallization without XPL imaging on thin-sections and/or cathodoluminescence? Crystal growth morphology is not well documented, so this is not entirely supported by the data provided. For example, Figure 3b and c may look like a single cementation phase with PPL but could show very different CL and XPL characteristics. I don't mean to be difficult here, but I have seen many examples that show this pattern.*

*Line 320 – The problem with multiple phases of growth is that they can occure at different stages of the diagenetic history. Also it is very hard to observe incremental growth in the provided images.*

The three above comments are all resolved by the addition of CL imaging to the paper, we would like to thank the reviewer for the suggestion as this additional petrological context helped with the interpretation of the results.

*Line 215 onward – I think the main limitation for the precision here is the primary reference material and not the secondary, as might be understood from this paragraph. The mapping technique is a real state-of-the-art approach; however, it is limited here by the used primary reference material (RMs - WC1), with a minimal uncertainty of 2-2.5%, thus, ages will be plus minus 0.5 Ma for an age of 20 Ma. This notion may help to push forward the use of better RMs in the future.*

A new paragraph about future work was added addressing this issue.

*Line 325 – I find the statistical explanation unnecessarily complicated, I think it can be simplified by first providing comparison for the different approaches such as age constraint base on the:*

1. *age with lowest uncertainties: 19.11 ±94 Ma*

2. *TW plots of all "spot" analyses : ???*

3. *radial plot: 19.34 ±73 Ma*

4. *mean weighted average : 19.32 ±73 Ma*

5. *oldest age, to account for post formation diagenetic processes: 20.4 ±6 Ma*

6. *Youngest age, to account for potential reworked contribution: 18 ±2 Ma*

*Then you can explain why you choose option number 3?*

We agree that this suggested outline is a better way to discuss the results, thus we modified this paragraph accordingly.

Response to reviewer: Andreas Möller – reviewer comments are in blue italics.

We would like to thank the reviewer for his comments, and all his suggestions were carefully incorporated with changes made to the text and figures.

*One of my main general points of criticism in terminology is the use of the term "direct dating" from the Title on throughout the manuscript. As the manuscript itself explains repeatedly, dating pedogenic nodules means dating a pedogenic process and not "directly" dating sedimentation. I understand that the difference may be within the uncertainty of the obtained dates, but it is still incorrect. I Therefore recommend strongly this phrase should not be used and be struck from the title. Even is nodule formation is happening shortly after deposition, it is not "direct".*

We concur that the term "direct dating" was probably used too loosely. While we date the calcite directly, it is true we do not directly date sedimentation but rather pedogenesis. To avoid confusion, we have reformulated this phrasing throughout.

*Introduction : Should take into account some recent paper on continental carbonates.*

We added relevant references to the introduction, including the one suggested in the comments.

*Results : The powder XRD data do not contribute significantly to the discussion or conclusion and could be put in the supplemental data. Chapter needs some checking of result and uncertainties to be matched between text and figures and an explanation why the two results from the same sample P00 are both used the average. This gives double weight to the result from that sample.*

The powder XRD data were removed and placed in the Supplementary Material. The misquoted age uncertainty was fixed and we checked all other uncertainties on figures and in the text to prevent any other discrepancies. The age results from the mapping experiment undertaken on sample P00 were removed from the statistical calculation and are only now used for comparison purposes, and only the spot analysis age results from sample P00 are now included in the statistical calculations.

*Discussion : Needs a bit more explanation in the biostratigraphic significance chapter, check the figure against the text, line 355ff. And be more specific how the approach used here is more reliable than others, line 362.*

A more detailed explanation based on both this comment and the suggestions in the annotated manuscript was added to the text and figures.

*Conclusions : The claims made here: "efficient" and "accurate" need to be better supported by specific explanations.*

We now give more elaboration regarding these points/claims, taking into consideration the comments raised by both reviewers.

*Figures :Some figures need more explanation or a better legend, Fig. 9 needs more explanation (see comment there).Supplementary figure 1 needs a detailed caption*

The relevant figures captions were revised according to the specific comments in the reviewer's annotated manuscript. Supplementary Figure 1 now has a detailed caption.

Tables : Supplemental Table 1 is missing from the supplementary files.

The analytical data is available at the following updated zenodo link:
https://doi.org/10.5281/zenodo.14500416
According to the instructions of *Geochronology* the dataset was referenced within the text using the hyperlink provided above.